# Inference by Learning: Speeding-up Graphical Model Optimization via a Coarse-to-Fine Cascade of Pruning Classifiers

**Bruno Conejo**[*]
GPS Division, California Institute of Technology, Pasadena, CA, USA
Universite Paris-Est, Ecole des Ponts ParisTech, Marne-la-Vallee, France
bconejo@caltech.edu

**Nikos Komodakis**
Universite Paris-Est, Ecole des Ponts ParisTech, Marne-la-Vallee, France
nikos.komodakis@enpc.fr

**Sebastien Leprince & Jean Philippe Avouac**
GPS Division, California Institute of Technology, Pasadena, CA, USA
leprincs@caltech.edu avouac@gps.caltech.edu

## Abstract

We propose a general and versatile framework that significantly speeds-up graphical model optimization while maintaining an excellent solution accuracy. The proposed approach, refereed as Inference by Learning or in short as IbyL, relies on a multi-scale pruning scheme that progressively reduces the solution space by use of a coarse-to-fine cascade of learnt classifiers. We thoroughly experiment with classic computer vision related MRF problems, where our novel framework constantly yields a significant time speed-up (with respect to the most efficient inference methods) and obtains a more accurate solution than directly optimizing the MRF. We make our code available on-line [4].

## 1 Introduction

**Graphical models in computer vision** Optimization of undirected graphical models such as Markov Random Fields, MRF, or Conditional Random Fields, CRF, is of fundamental importance in computer vision. Currently, a wide spectrum of problems including stereo matching [25, 13], optical flow estimation [27, 16], image segmentation [23, 14], image completion and denoising [10], or, object recognition [8, 2] rely on finding the mode of the distribution associated to the random field, i.e., the Maximum A Posteriori (MAP) solution. The MAP estimation, often referred as the labeling problem, is posed as an energy minimization task. While this task is NP-Hard, strong optimum solutions or even the optimal solutions can be obtained [3]. Over the past 20 years, tremendous progress has been made in term of computational cost, and, many different techniques have been developed such as move making approaches [3, 19, 22, 21, 28], and message passing methods [9, 32, 18, 20]. A review of their effectiveness has been published in [31, 12]. Nevertheless, the ever increasing dimensionality of the problems and the need for larger solution space greatly challenge these tech-

---

[*]This work was supported by USGS through the Measurements of surface ruptures produced by continental earthquakes from optical imagery and LiDAR project (USGS Award G13AP00037), the Terrestrial Hazard Observation and Reporting Center of Caltech, and the Moore foundation through the Advanced Earth Surface Observation Project (AESOP Grant 2808).

niques as even the best ones have a highly super-linear computational cost and memory requirement relatively to the dimensionality of the problem.

Our goal in this work is to develop a general MRF optimization framework that can provide a significant speed-up for such methods while maintaining the accuracy of the estimated solutions. Our strategy for accomplishing this goal will be to gradually reduce (by a significant amount) the size of the discrete state space via exploiting the fact that an optimal labeling is typically far from being random. Indeed, most MRF optimization problems favor solutions that are piecewise smooth. In fact, this spatial structure of the MAP solution has already been exploited in prior work to reduce the dimensionality of the solution space.

**Related work**    A first set of methods of this type, referred here for short as the *super-pixel* approach [30], defines a grouping heuristic to merge many random variables together in super-pixels. The grouping heuristic can be energy-aware if it is based on the energy to minimize as in [15], or, energy-agnostic otherwise as in [7, 30]. All random variables belonging to the same super-pixel are forced to take the same label. This restricts the solution space and results in an optimization speed-up as a smaller number of variables needs to be optimized. The super-pixel approach has been applied with segmentation, stereo and object recognition [15]. However, if the grouping heuristic merges variables that should have a different label in the MAP solution, only an approximate labeling is computed. In practice, defining general yet efficient grouping heuristics is difficult. This represents the key limitation of super-pixel approaches.

One way to overcome this limitation is to mimic the multi-scale scheme used in continuous optimization by building a coarse to fine representation of the graphical model. Similarly to the super-pixel approach, such a *multi-scale* method, relies again on a grouping of variables for building the required coarse to fine representation [17, 24, 26]. However, contrary to the super-pixel approach, if the grouping merges variables that should have a different label in the MAP solution, there always exists a scale at which these variables are not grouped. This property thus ensures that the MAP solution can still be recovered. Nevertheless, in order to manage a significant speed-up of the optimization, the multi-scale approach also needs to progressively reduce the number of labels per random variable (i.e., the solution space). Typically, this is achieved by use, for instance, of a heuristic that keeps only a small fixed number of labels around the optimal label of each node found at the current scale, while pruning all other labels, which are therefore not considered thereafter [5]. This strategy, however, may not be optimal or even valid for all types of problems. Furthermore, such a pruning heuristic is totally inappropriate (and can thus lead to errors) for nodes located along discontinuity boundaries of an optimal solution, where such boundaries are always expected to exist in practice. An alternative strategy followed by some other methods relies on selecting a subset of the MRF nodes at each scale (based on some criterion) and then fixing their labels according to the optimal solution estimated at the current scale (essentially, such methods contract the entire label set of a node to a single label). However, such a fixing strategy may be too aggressive and can also easily lead to eliminating good labels.

**Proposed approach**    Our method simultaneously makes use of the following two strategies for speeding-up the MRF optimization process:

(i) it solves the problem through a multi-scale approach that gradually refines the MAP estimation based on a coarse-to-fine representation of the graphical model,

(ii) and, at the same time, it progressively reduces the label space of each variable by cleverly utilizing the information computed during the above coarse-to-fine process.

To achieve that, we propose to significantly revisit the way that the pruning of the solution space takes place. More specifically:

(i) we make use of and incorporate into the above process a *fine-grained* pruning scheme that allows an arbitrary subset of labels to be discarded, where this subset can be different for each node,

(ii) additionally, and most importantly, instead of trying to manually come up with some criteria for deciding what labels to prune or keep, we introduce the idea of relying entirely on a sequence of *trained classifiers* for taking such decisions, where *different classifiers per scale* are used.

We name such an approach *Inference by Learning*, and show that it is particularly *efficient* and *effective* in reducing the label space while omitting very few correct labels. Furthermore, we demonstrate that the training of these classifiers can be done based on features that are not application specific but depend solely on the energy function, which thus makes our approach generic and applicable to any MRF problem. The end result of this process is to obtain both an important speed-up and a significant decrease in memory consumption as the solution space is progressively reduced. Furthermore, as each scale refines the MAP estimation, a further speed-up is obtained as a result of a warm-start initialization that can be used when transitioning between different scales.

Before proceeding, it is worth also noting that there exists a body of prior work [29] that focuses on fixing the labels of a subset of nodes of the graphical model by searching for a partial labeling with the so-called *persistency* property (which means that this labeling is provably guaranteed to be part of an optimal solution). However, finding such a set of persistent variables is typically very time consuming. Furthermore, in many cases only a limited number of these variables can be detected. As a result, the focus of these works is entirely different from ours, since the main motivation in our case is how to obtain a significant speed-up for the optimization.

Hereafter, we assume without loss of generality that the graphical model is a discrete pairwise CRF/MRF. However, one can straightforwardly apply our approach to higher order models.

**Outline of the paper**  We briefly review the optimization problem related to a discrete pairwise MRF and introduce the necessary notations in section 2. We describe our general multi-scale pruning framework in section 3. We explain how classifiers are trained in section 4. Experimental results and their analysis are presented in 5. Finally, we conclude the paper in section 6.

## 2   Notation and preliminaries

To represent a discrete MRF model $\mathcal{M}$, we use the following notation

$$\mathcal{M} = \left( \mathcal{V}, \mathcal{E}, \mathcal{L}, \{\phi_i\}_{i \in \mathcal{V}}, \{\phi_{ij}\}_{(i,j) \in \mathcal{E}} \right). \tag{1}$$

Here $\mathcal{V}$ and $\mathcal{E}$ represent respectively the nodes and edges of a graph, and $\mathcal{L}$ represents a discrete label set. Furthermore, for every $i \in \mathcal{V}$ and $(i,j) \in \mathcal{E}$, the functions $\phi_i : \mathcal{L} \to \mathbb{R}$ and $\phi_{ij} : \mathcal{L}^2 \to \mathbb{R}$ represent respectively unary and pairwise costs (that are also known connectively as MRF potentials $\phi = \left\{ \{\phi_i\}_{i \in \mathcal{V}}, \{\phi_{ij}\}_{(i,j) \in \mathcal{E}} \right\}$). A solution $x = (x_i)_{i \in \mathcal{V}}$ of this model consists of one variable per vertex $i$, taking values in the label set $\mathcal{L}$, and the total cost (energy) $E(x|\mathcal{M})$ of such a solution is

$$E(x|\mathcal{M}) = \sum_{i \in V} \phi_i(x_i) + \sum_{(i,j) \in \mathcal{E}} \phi_{ij}(x_i, x_j) \ .$$

The goal of MAP estimation is to find a solution that has minimum energy, i.e., computes

$$x_{\mathrm{MAP}} = \arg \min_{x \in \mathcal{L}^{|\mathcal{V}|}} E(x|\mathcal{M}) \ .$$

The above minimization takes place over the full solution space of model $\mathcal{M}$, which is $\mathcal{L}^{|\mathcal{V}|}$. Here we will also make use of a pruned solution space $\mathcal{S}(\mathcal{M}, A)$, which is defined based on a binary function $A : \mathcal{V} \times \mathcal{L} \to \{0, 1\}$ (referred to as the *pruning matrix* hereafter) that specifies the status (active or pruned) of a label for a given vertex, i.e.,

$$A(i, l) = \left\{ \begin{array}{ll} 1 & \text{if label } l \text{ is active at vertex } i \\ 0 & \text{if label } l \text{ is pruned at vertex } i \end{array} \right. \tag{2}$$

During optimization, active labels are retained while pruned labels are discarded. Based on a given $A$, the corresponding pruned solution space of model $\mathcal{M}$ is defined as

$$\mathcal{S}(\mathcal{M}, A) = \left\{ x \in \mathcal{L}^{|\mathcal{V}|} \mid (\forall i), A(i, x_i) = 1 \right\} \ .$$

## 3   Multiscale Inference by Learning

In this section we describe the overall structure of our MAP estimation framework, beginning by explaining how to construct the coarse-to-fine representation of the input graphical model.

## 3.1 Model coarsening

Given a model $\mathcal{M}$ (defined as in (1)), we wish to create a "coarser" version of this model $\mathcal{M}' = \left( \mathcal{V}', \mathcal{E}', \mathcal{L}, \{\phi_i'\}_{i \in \mathcal{V}'}, \{\phi_{ij}'\}_{(i,j) \in \mathcal{E}'} \right)$. Intuitively, we want to partition the nodes of $\mathcal{M}$ into groups, and treat each group as a single node of the coarser model $\mathcal{M}'$ (the implicit assumption is that nodes of $\mathcal{M}$ that are grouped together are assigned the same label). To that end, we will make use of a grouping function $g : \mathcal{V} \to \mathcal{N}$. The nodes and edges of the coarser model are then defined as follows

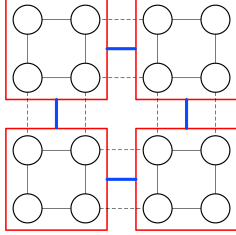

$$\mathcal{V}' = \{i' \mid \exists i \in \mathcal{V}, i' = g(i)\} , \tag{3}$$

$$\mathcal{E}' = \{(i', j') \mid \exists (i,j) \in \mathcal{E}, i' = g(i), j' = g(j), i' \neq j'\} . \tag{4}$$

Furthermore, the unary and pairwise potentials of $\mathcal{M}'$ are given by

$$(\forall i' \in \mathcal{V}'), \quad \phi_{i'}'(l) = \sum_{i \in \mathcal{V} \mid i' = g(i)} \phi_i(l) + \sum_{(i,j) \in \mathcal{E} \mid i' = g(i) = g(j)} \phi_{ij}(l,l) , \tag{5}$$

$$(\forall (i', j') \in \mathcal{E}'), \quad \phi_{i'j'}'(l_0, l_1) = \sum_{(i,j) \in \mathcal{E} \mid i' = g(i), j' = g(j)} \phi_{ij}(l_0, l_1) . \tag{6}$$

Figure 1: Black circles: $\mathcal{V}$, Black lines: $\mathcal{E}$, Red squares: $\mathcal{V}'$, Blue lines: $\mathcal{E}'$.

With a slight abuse of notation, we will hereafter use $g(\mathcal{M})$ to denote the coarser model resulting from $\mathcal{M}$ when using the grouping function $g$, i.e., we define $g(\mathcal{M}) = \mathcal{M}'$. Also, given a solution $x'$ of $\mathcal{M}'$, we can "upsample" it into a solution $x$ of $\mathcal{M}$ by setting $x_i = x'_{g(i)}$ for each $i \in \mathcal{V}$. We will use the following notation in this case: $g^{-1}(x') = x$. We provide a toy example in supplementary materials.

## 3.2 Coarse-to-fine optimization and label pruning

To estimate the MAP of an input model $\mathcal{M}$, we first construct a series of $N+1$ progressively coarser models $(\mathcal{M}^{(s)})_{0 \leq s \leq N}$ by use of a sequence of $N$ grouping functions $(g^{(s)})_{0 \leq s < N}$, where

$$\mathcal{M}^{(0)} = \mathcal{M} \quad \text{and} \quad (\forall s), \ \mathcal{M}^{(s+1)} = g^{(s)}(\mathcal{M}^{(s)}) .$$

This provides a multiscale (coarse-to-fine) representation of the original model., where the elements of the resulting models are denoted as follows:

$$\mathcal{M}^{(s)} = \left( \mathcal{V}^{(s)}, \mathcal{E}^{(s)}, \mathcal{L}, \{\phi_i^{(s)}\}_{i \in \mathcal{V}^{(s)}}, \{\phi_{ij}^{(s)}\}_{(i,j) \in \mathcal{E}^{(s)}} \right)$$

In our framework, MAP estimation proceeds from the coarsest to the finest scale (i.e., from model $\mathcal{M}^{(N)}$ to $\mathcal{M}^{(0)}$). During this process, a pruning matrix $A^{(s)}$ is computed at each scale $s$, which is used for defining a restricted solution space $\mathcal{S}(\mathcal{M}^{(s)}, A^{(s)})$. The elements of the matrix $A^{(N)}$ at the coarsest scale are all set equal to 1 (i.e., no label pruning is used in this case), whereas in all other scales $A^{(s)}$ is computed by use of a trained classifier $f^{(s)}$.

More specifically, at any given scale $s$, the following steps take place:

i. We approximately minimize (via any existing MRF optimization method) the energy of the model $\mathcal{M}^{(s)}$ over the restricted solution space $\mathcal{S}(\mathcal{M}^{(s)}, A^{(s)})$, i.e., we compute

$$x^{(s)} \approx \arg\min_{x \in \mathcal{S}(\mathcal{M}^{(s)}, A^{(s)})} E(x | \mathcal{M}^{(s)}) .$$

ii. Given the estimated solution $x^{(s)}$, a feature map $z^{(s)} : \mathcal{V}^{(s)} \times \mathcal{L} \to \mathbb{R}^K$ is computed at the current scale, and a trained classifier $f^{(s)} : \mathbb{R}^K \to \{0, 1\}$ uses this feature map $z^{(s)}$ to construct the pruning matrix $A^{(s-1)}$ for the next scale as follows

$$(\forall i \in \mathcal{V}^{(s-1)}, \ \forall l \in \mathcal{L}), \quad A^{(s-1)}(i, l) = f^{(s)}(z^{(s)}(g^{(s-1)}(i), l)) .$$

iii. Solution $x^{(s)}$ is "upsampled" into $x^{(s-1)} = [g^{(s-1)}]^{-1}(x^{(s)})$ and used as the initialization for the optimization at the next scale $s - 1$. Note that, due to (5) and (6), it holds $E(x^{(s-1)} | \mathcal{M}^{(s-1)}) = E(x^{(s)} | \mathcal{M}^{(s)})$. Therefore, this initialization ensures that energy will continually decrease if the same is true for the optimization applied per scale. Furthermore, it can allow for a warm-starting strategy when transitioning between scales.

The pseudocode of the resulting algorithm appears in Algo. 1.

**Algorithm 1:** Inference by learning framework

---

**Data**: Model $\mathcal{M}$, grouping functions $(g^{(s)})_{0 \leq s < N}$, classifiers $(f^{(s)})_{0 < s \leq N}$
**Result**: $x^{(0)}$
Compute the coarse to fine sequence of MRFs:
$\mathcal{M}^{(0)} \leftarrow \mathcal{M}$
**for** $s = [0 \ldots N - 1]$ **do**
$\quad \lfloor \; \mathcal{M}^{(s+1)} \leftarrow g^{(s)}(\mathcal{M}^{(s)})$
Optimize the coarse to fine sequence of MRFs over pruned solution spaces:
$(\forall i \in \mathcal{V}^{(N)}, \forall l \in \mathcal{L})$, $A^{(N)}(i, l) \leftarrow 1$
Initialize $x^{(N)}$
**for** $s = [N \ldots 0]$ **do**
$\quad$ Update $x^{(s)}$ by iterative minimization: $x^{(s)} \approx \arg\min_{x \in \mathcal{S}(\mathcal{M}^{(s)}, A^{(s)})} E(x | \mathcal{M}^{(s)})$
$\quad$ **if** $s \neq 0$ **then**
$\quad\quad$ Compute feature map $z^{(s)}$
$\quad\quad$ Update pruning matrix for next finer scale: $A^{(s-1)}(i, l) = f^{(s)}(z^{(s)}(g^{(s-1)}(i), l))$
$\quad\quad$ Upsample $x^{(s)}$ for initializing solution $x^{(s-1)}$ at next scale: $x^{(s-1)} \leftarrow [g^{(s-1)}]^{-1}(x^{(s)})$

---

## 4 Features and classifier for label pruning

For each scale $s$, we explain how the set of features comprising the feature map $z^{(s)}$ is computed and how we train (off-line) the classifier $f^{(s)}$. This is a crucial step for our approach. Indeed, if the classifier wrongly prunes labels that belong to the MAP solution, then, only an approximate labeling might be found at the finest scale. Moreover, keeping too many active labels will result in a poor speed-up for MAP estimation.

### 4.1 Features

The feature map $z^{(s)} : \mathcal{V}^{(s)} \times \mathcal{L} \to \mathbb{R}^K$ is formed by stacking $K$ individual real-valued features defined on $\mathcal{V}^{(s)} \times \mathcal{L}$. We propose to compute features that are not application specific but depend solely on the energy function and the current solution $x^{(s)}$. This makes our approach generic and applicable to any MRF problem. However, as we establish a general framework, specific application features can be straightforwardly added in future work.

**Presence of strong discontinuity** This binary feature, $\mathrm{PSD}^{(s)}$, accounts for the existence of discontinuity in solution $x^{(s)}$ when a strong link (i.e., $\phi_{ij}(x_i^{(s)}, x_j^{(s)}) > \rho$) exists between neighbors. Its definition follows for any vertex $i \in \mathcal{V}^{(s)}$ and any label $l \in \mathcal{L}$ :

$$\mathrm{PSD}^{(s)}(i, l) = \begin{cases} 1 & \exists (i, j) \in \mathcal{E}^{(s)} | \; \phi_{ij}(x_i^{(s)}, x_j^{(s)}) > \rho \\ 0 & \text{otherwise} \end{cases} \tag{7}$$

**Local energy variation** This feature represents the local variation of the energy around the current solution $x^{(s)}$. It accounts for both the unary and pairwise terms associated to a vertex and a label. As in [11], we remove the local energy of the current solution as it leads to a higher discriminative power. The local energy variation feature, $\mathrm{LEV}^{(s)}$, is defined for any $i \in \mathcal{V}^{(s)}$ and $l \in \mathcal{L}$ as follows:

$$\mathrm{LEV}^{(s)}(i, l) = \frac{\phi_i^{(s)}(l) - \phi_i^{(s)}(x_i^{(s)})}{N_{\mathcal{V}}^{(s)}(i)} + \sum_{j:(i,j)\in\mathcal{E}^{(s)}} \frac{\phi_{ij}^{(s)}(l, x_j^{(s)}) - \phi_{ij}^{(s)}(x_i^{(s)}, x_j^{(s)})}{N_{\mathcal{E}}^{(s)}(i)} \tag{8}$$

with $N_{\mathcal{V}}^{(s)}(i) = \mathrm{card}\{i' \in \mathcal{V}^{(s-1)} : g^{(s-1)}(i') = i\}$ and $N_{\mathcal{E}}^{(s)}(i) = \mathrm{card}\{(i', j') \in \mathcal{E}^{(s-1)} : g^{(s-1)}(i') = i, g^{(s-1)}(j') = j\}$.

**Unary "coarsening"** This feature, $\mathrm{UC}^{(s)}$, aims to estimate an approximation of the coarsening induced in the MRF unary terms when going from model $\mathcal{M}^{(s-1)}$ to model $\mathcal{M}^{(s)}$, i.e., as a result of

applying the grouping function $g^{(s-1)}$. It is defined for any $i \in \mathcal{V}^{(s)}$ and $l \in \mathcal{L}$ as follows

$$\text{UC}^{(s)}(i,l) = \sum_{i' \in \mathcal{V}^{(s-1)} | g^{(s-1)}(i')=i} \frac{|\phi_{i'}^{(s-1)}(l) - \frac{\phi_i^{(s)}(l)}{N_\mathcal{V}^{(s)}(i)}|}{N_\mathcal{V}^{(s)}(i)} \tag{9}$$

**Feature normalization**   The features are by design insensitive to any additive term applied on all the unary and pairwise terms. However, we still need to apply a normalization to the $\text{LEV}^{(s)}$ and $\text{UC}^{(s)}$ features to make them insensitive to any positive global scaling factor applied on both the unary and pairwise terms (such scaling variations are commonly used in computer vision). Hence, we simply divide group of features, $\text{LEV}^{(s)}$ and $\text{UC}^{(s)}$ by their respective mean value.

### 4.2   Classifier

To train the classifiers, we are given as input a set of MRF instances (all of the same class, e.g., stereo-matching) along with the ground truth MAP solutions. We extract a subset of MRFs for off-line learning and a subset for on-line testing. For each MRF instance in the training set, we apply the algorithm 1 without any pruning (i.e., $A^{(s)} \equiv 1$) and, at each scale, we keep track of the features $z^{(s)}$ and also compute the binary function $X_{\text{MAP}}^{(s)} : \mathcal{V}^{(s)} \times \mathcal{L} \to \{0,1\}$ defined as follows:

$$(\forall i \in \mathcal{V}, \forall l \in \mathcal{L}), \quad X_{\text{MAP}}^{(0)}(i,l) = \begin{cases} 1, & \text{if } l \text{ is the ground truth label for node } i \\ 0, & \text{otherwise} \end{cases}$$

$$(\forall s > 0)(\forall i \in \mathcal{V}^{(s)}, \forall l \in \mathcal{L}), \quad X_{\text{MAP}}^{(s)}(i,l) = \bigvee_{i' \in \mathcal{V}^{(s-1)} : g^{(s)}(i')=i} X_{\text{MAP}}^{(s-1)}(i',l) ,$$

where $\bigvee$ denotes the binary OR operator. The values 0 and 1 in $X_{\text{MAP}}^{(s)}$ define respectively the two classes $c_0$ and $c_1$ when training the classifier $f^{(s)}$, where $c_0$ means that the label can be pruned and $c_1$ that the label should not be pruned.

To treat separately the nodes that are on the border of a strong discontinuity, we split the feature map $z^{(s)}$ into two groups $z_0^{(s)}$ and $z_1^{(s)}$, where $z_0^{(s)}$ contains only features where $\text{PSD}^{(s)} = 0$ and $z_1^{(s)}$ contains only features where $\text{PSD}^{(s)} = 1$ (strong discontinuity). For each group, we train a standard linear C-SVM classifier with $l_2$-norm regularization (regularization parameter was set to $C = 10$). The linear classifiers give good enough accuracy during training while also being fast to evaluate at test time

During training (and for each group), we also introduce weights to balance the different number of elements in each class ($c_0$ is much larger than $c_1$), and to also strongly penalize misclassification in $c_1$ (as such misclassification can have a more drastic impact on the accuracy of MAP estimation). To accomplish that, we set the weight for class $c_0$ to 1, and the weight for class $c_1$ to $\lambda \frac{\text{card}(c_0)}{\text{card}(c_1)}$, where $\text{card}(\cdot)$ counts the number of training samples in each class. Parameter $\lambda$ is a positive scalar (common to both groups) used for tuning the penalization of misclassification in $c_1$ (it will be referred to as the pruning aggressiveness factor hereafter as it affects the amount of labels that get pruned). During on-line testing, depending on the value of the PSD feature, $f^{(s)}$ applies the linear classifier learned on group $z_0^{(s)}$ if $\text{PSD}^{(s)} = 0$, or the linear classifier learned on group $z_1^{(s)}$ if $\text{PSD}^{(s)} = 1$.

## 5   Experimental results

We evaluate our framework on pairwise MRFs from stereo-matching, image restoration, and, optical flow estimation problems. The corresponding MRF graphs consist of regular 4-connected grids in this case. At each scale, the grouping function merges together vertices of $2 \times 2$ subgrids. We leave more advanced grouping functions [15] for future work. As MRF optimization subroutine, we use the Fast-PD algorithm [21]. We make our code available on-line [4].

**Experimental setup**   For the stereo matching problem, we estimate the disparity map from images $I_R$ and $I_L$ where each label encodes a potential disparity $d$ (discretized at quarter of a pixel precision), with MRF potentials $\phi_p(d) = ||I_L(y_p, x_p) - I_R(y_p, x_p - d)||_1$ and $\phi_{pq}(d_0, d_1) = w_{pq}|d_0 - d_1|$, with the weight $w_{pq}$ varying based on the image gradient (parameters are adjusted for each sequence). We train the classifier on the well-known *Tsukuba* stereo-pair (61 labels), and use all other

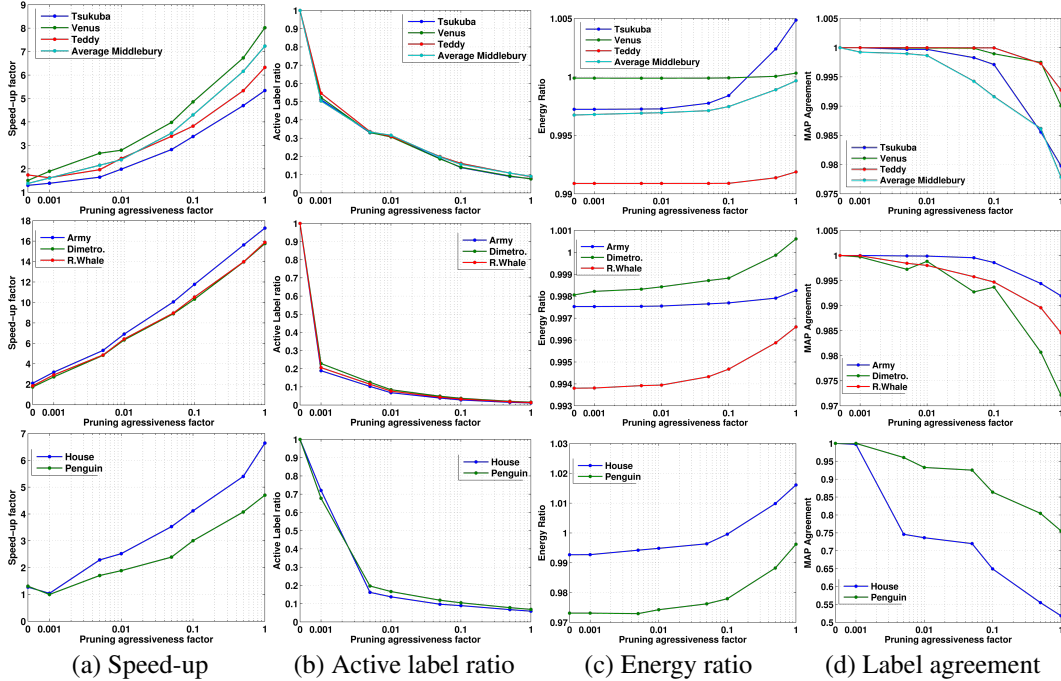

| (a) Speed-up | (b) Active label ratio | (c) Energy ratio | (d) Label agreement |

Figure 2: Performance of our Inference by Learning framework: (Top row) stereo matching, (Middle row) optical flow, (Bottom row) image restoration. For stereo matching, the Average Middlebury curve represents the average value of the statistic for the entire Middlebury dataset [6] (2001, 2003, 2005 and 2006) (37 stereo-pairs).

stereo-pairs of [6] (2001, 2003, 2005 and 2006) for testing. For image restoration, we estimate the pixel intensity of a noisy and incomplete image $I$ with MRF potentials $\phi_p(l) = ||I(y_p, x_p) - l||_2^2$ and $\phi(l_0, l_1) = 25 \min(||l_0 - l_1||_2^2, 200)$. We train the classifier on the *Penguin* image stereo-pair (256 labels), and use *House* (256 labels) for testing (dataset [31]). For the optical flow estimation, we estimate a subpixel-accurate 2D displacement field between two frames by extending the stereo matching formulation to 2D. Using the dataset of [1], we train the classifier on *Army* (1116 labels), and test on *RubberWhale* (625 labels) and *Dimetrodon* (483 labels). For all experiments, we use 5 scales and set in (7) $\rho = 5\bar{w}_{pq}$ with $\bar{w}_{pq}$ being the mean value of edge weights.

**Evaluations**  We evaluate three optimization strategies: the direct optimization (i.e., optimizing the full MRF at the finest scale), the multi-scale optimization ($\lambda = 0$, i.e., our framework without any pruning), and our Inference by Learning optimization, where we experiment with different error ratios $\lambda$ that range between $0.001$ and $1$.

We assess the performance by computing the *energy ratio*, i.e., the ratio between the current energy and the energy computed by the direct optimization, the *best label agreement*, i.e., the proportion of labels that coincides with the labels of the lowest computed energy, the *speed-up factor*, i.e., the ratio of computation time between the direct optimization and the current optimization strategy, and, the *active label ratio*, i.e., the percentage of active labels at the finest scale.

**Results and discussion**  For all problems, we present in Fig. 2 the performance of our Inference by Learning approach for all tested aggressiveness factors and show in Fig. 3 estimated results for $\lambda = 0.01$. We present additional results in the supplementary material.

For every problem and aggressiveness factors until $\lambda = 0.1$, our pruning-based optimization obtains a lower energy (column (c) of Fig. 2) in less computation time, achieving a speed-up factor (column (a) of Fig. 2) close to 5 for Stereo-matching, above 10 for Optical-flow and up to 3 for image restoration. (note that these speed-up factors are with respect to an algorithm, FastPD, that was the most efficient one in recent comparisons [12]). The percentage of active labels (Fig. 2 column (b)) strongly correlates with the speed-up factor. The best labeling agreement (Fig. 2 column (d)) is never worse than $97\%$ (except for the image restoration problems because of the in-painted area)

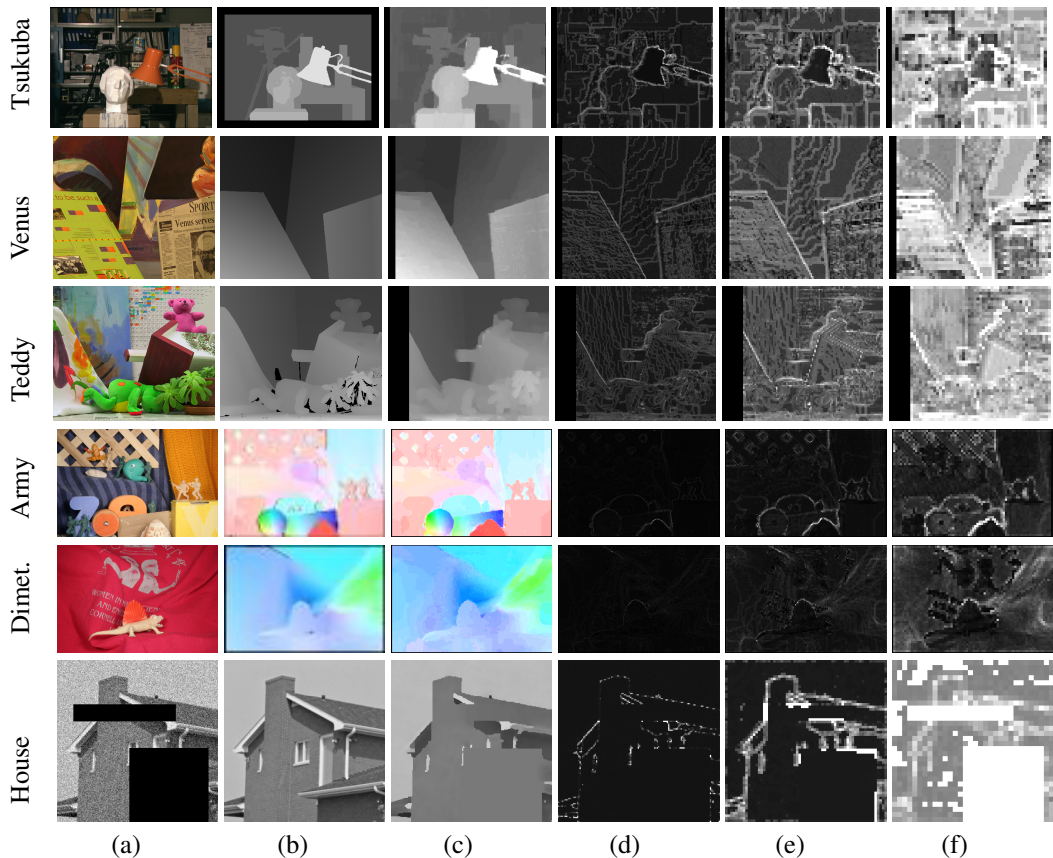

Figure 3: Results of our Inference by Learning framework for $\lambda = 0.1$. Each row is a different MRF problem. (a) original image, (b) ground truth, (c) solution of the pruning framework, (d,e,f) percentage of active labels per vertex for scale 0, 1 and 2 (black 0%, white 100%).

and is always above 99% for $\lambda \leqslant 0.1$. As expected, less pruning happens near label discontinuities as illustrated in column (d,e,f) of Fig. 3 justifying the use of a dedicated linear classifier. Moreover, large homogeneously labeled regions are pruned earlier in the coarse to fine scale.

## 6   Conclusion and future work

Our *Inference by Learning* approach consistently speeds-up the graphical model optimization by a significant amount while maintaining an excellent accuracy of the labeling estimation. On most experiments, it even obtains a lower energy than direct optimization.

In future work, we plan to experiment with problems that require general pairwise potentials where message-passing techniques can be more effective than graph-cut based methods but are at the same time much slower. Our framework is guaranteed to provide an even more dramatic speedup in this case since the computational complexity of message-passing methods is quadratic with respect to the number of labels while being linear for graph-cut based methods used in our experiments. We also intend to explore the use of application specific features, learn the grouping functions used in the coarse-to-fine scheme, jointly train the cascade of classifiers, and apply our framework to high order graphical models.

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
