[Supplementary Material]

# Inference by Learning: Speeding-up Graphical Model Optimization via a Coarse-to-Fine Cascade of Pruning Classifiers
# -
# Supplemental materials

**Bruno Conejo**[*]
GPS Division, California Institute of Technology, Pasadena, CA, USA
Universite Paris-Est, Ecole des Ponts ParisTech, Marne-la-Vallee, France
`bconejo@caltech.edu`

**Nikos Komodakis**
Universite Paris-Est, Ecole des Ponts ParisTech, Marne-la-Vallee, France
`nikos.komodakis@enpc.fr`

**Sebastien Leprince & Jean Philippe Avouac**
GPS Division, California Institute of Technology, Pasadena, CA, USA
`leprincs@caltech.edu avouac@gps.caltech.edu`

## A   Toy example for model coarsening

Given $\mathcal{M} = \big(\mathcal{V}, \mathcal{E}, \mathcal{L}, \{\phi_i\}_{i\in\mathcal{V}}, \{\phi_{ij}\}_{(i,j)\in\mathcal{E}}\big)$, a graphical model, we create the coarsened graphical model $\mathcal{M}' = \big(\mathcal{V}', \mathcal{E}', \mathcal{L}, \{\phi_i'\}_{i\in\mathcal{V}'}, \{\phi_{ij}'\}_{(i,j)\in\mathcal{E}'}\big)$.

The vertices of $\mathcal{V}$ are the black circles and the the black thin solid and dashed lines are the edges $\mathcal{E}$. In this toy example, we consider a grouping function $g$ that merges vertices of $\mathcal{V}$ of $2 \times 2$ subgrids together.

The grouping function $g$ creates the coarsened graphical model $\mathcal{M}'$ where the red squares are the new induced vertices $\mathcal{V}'$ and the solid thick blue lines are the new induced edges $\mathcal{E}'$.

Figure 1: Toy example for model coarsening

---
[*]This work was supported by USGS through the Measurements of surface ruptures produced by continental earthquakes from optical imagery and LiDAR project (USGS Award G13AP00037), the Terrestrial Hazard Observation and Reporting Center of Caltech, and the Moore foundation through the Advanced Earth Surface Observation Project (AESOP Grant 2808).

# B Results

## B.1 Stereo

### B.1.1 Tsukuba

Figure 2: Results of our Inference by Learning framework for Tsukuba. Each row is a different pruning aggressiveness value ($\lambda$). (a) original image, (b) ground truth, (c) solution of the pruning framework, (d,e,f) percentage of active labels per vertex for scale 0, 1 and 2 (black $0\%$, white $100\%$).

### B.1.2 Venus

Figure 3: Results of our Inference by Learning framework for Venus. Each row is a different pruning aggressiveness value ($\lambda$). (a) original image, (b) ground truth, (c) solution of the pruning framework, (d,e,f) percentage of active labels per vertex for scale 0, 1 and 2 (black $0\%$, white $100\%$).

### B.1.3 Teddy

Figure 4: Results of our Inference by Learning framework for Teddy. Each row is a different pruning aggressiveness value ($\lambda$). (a) original image, (b) ground truth, (c) solution of the pruning framework, (d,e,f) percentage of active labels per vertex for scale 0, 1 and 2 (black 0%, white 100%).

## B.2 Optical-Flow

### B.2.1 Army

Figure 5: Results of our Inference by Learning framework for Army. Each row is a different pruning aggressiveness value ($\lambda$). (a) original image, (b) ground truth, (c) solution of the pruning framework, (d,e,f) percentage of active labels per vertex for scale 0, 1 and 2 (black 0%, white 100%).

### B.2.2  Dimetrodon

Figure 6: Results of our Inference by Learning framework for Dimetrodon. Each row is a different pruning aggressiveness value ($\lambda$). (a) original image, (b) ground truth, (c) solution of the pruning framework, (d,e,f) percentage of active labels per vertex for scale 0, 1 and 2 (black 0%, white 100%).

### B.2.3  Rubberwhale

Figure 7: Results of our Inference by Learning framework for Rubberwhale. Each row is a different pruning aggressiveness value ($\lambda$). (a) original image, (b) ground truth, (c) solution of the pruning framework, (d,e,f) percentage of active labels per vertex for scale 0, 1 and 2 (black 0%, white 100%).

## B.3 Denoising
### B.3.1 Penguin

Figure 8: Results of our Inference by Learning framework for Penguin. Each row is a different pruning aggressiveness value ($\lambda$). (a) original image, (b) ground truth, (c) solution of the pruning framework, (d,e,f) percentage of active labels per vertex for scale 0, 1 and 2 (black $0\%$, white $100\%$).

## B.3.2 House

Figure 9: Results of our Inference by Learning framework for House. Each row is a different pruning aggressiveness value ($\lambda$). (a) original image, (b) ground truth, (c) solution of the pruning framework, (d,e,f) percentage of active labels per vertex for scale 0, 1 and 2 (black $0\%$, white $100\%$).