[Reviews · NeurIPS 2014]

Submitted by Assigned_Reviewer_8

The paper is aiming to speed up optimization for the of graphical model MAP problem. The authors suggest a coarse to fine approach and clearly motivate their work. The method suggested by the authors combines label pruning for each variable with sequential optimization of the pruned, yet finer problem.

As far as i can tell, the work is original and correct. One point i think was not clearly pointed out is whether the proposed method can be applied to graph structure with no apparent regularity (in contrast to grid graphs resulting from images).

Another interesting question i am curious about is whether this method could be used inside a learning procedure.

Given my limited knowledge of the related works it is hard for me to asses the significance of this work.
Summary: The papers proposes a coarse to fine method to accelerate MAP inference. The method seems interesting and sound,. In addition experimental result are also supportive.

Submitted by Assigned_Reviewer_46

The manuscript “Speeding-up Graphical Model Optimization via a Coarse-to-fine Cascade of Pruning Classifiers” introduces a framework to improve MRF Optimization through a cascade of classifiers to progressively prune the search space while going from coarse to fine-grained models.

The newly introduced pruning scheme is novel, interesting and seems to provide good solution in practice while being faster than other approaches. There are several parameters pre-set (e.g., C = 10, 5 scales, rho). It is not clear, how robust the method is regarding the value of the parameters, but since they are set the same way for all evaluation data sets, they are probably not too over-trained (although line 294 states that C was set to 10, since it gave good enough accuracy (implying that the value was found looking at the test data)).

The manuscript is clearly written and easily accessible. The speed-up of the new method is considerable which is why it could have a significant impact on the field.

Minor typos:
Line 90: different classifiers per scale are used (not “is used”)
Lines 102-103: with a fine-grained pruning of the labels (not “fined-grained”)
Line 155: one with too much in the heading
Line 363: all aggressiveness factors except lambda (not “expect lambda”)
Line 370: illustrated in column (d,e,f) (not “(d,e,g)”)
Summary: The newly introduced pruning scheme is novel, interesting and seems to provide good solution in practice while being faster than other approaches. The speed-up of the new method is considerable which is why it could have a significant impact on the field.

Submitted by Assigned_Reviewer_48

This paper combines a coarse-to-fine cascade with pruning classifiers to speed up the optimization of the MRF.

Although the greedy method that is used to prune the solution space seems interesting, the paper does not have sufficient theoretical and empirical evidence to support the argument that the heuristic works. Also, the type of MRFs which may benefit from the method is limited to the ones whose solutions are piecewise smooth.

Some part of the paper is not clear enough. For example, in the description of the pruning matrix, it is not clear enough what "active" means. The pruning matrix updating of the algorithm is also not very clear. I would suggest the author provide a toy example to illustrate the algorithm better. On the other hand, the proposed approach part of sec 1 is too long and some contents may be redundant. Figure 2 is also not very informative.

The experiment is interesting but also not clear enough. First of all, there are three algorithms which are compared, but there is only one curve shown in fig 1 (Since there are three algorithms in comparison, only showing the ratio is not sufficient). Secondly, since the energy ratio is between the current energy and the lowest computed energy by any strategy, why is it less than 1 in many cases?

** After author feedback:
The authors clarify some of the confusions in the original paper. I expect the authors to correct all the typos and address all the concerns by the reviewers. I move my rating from 4 to 5, because there is still no theoretical justification for the pruning method.
A few more comments about the experiments:
- In column(d) of fig 1, both two curves in the 3rd row have less than 96% agreement, while in the paper it says "the agreement is never worse than 96%".
- Consider moving fig 2 to appendix. Add more experiments or analysis on hyperparameter choosing.
Summary: Interesting heuristics. Lack of clarity. Weak in theoretical and empirical support.
Author Feedback
Author rebuttal: We thank all reviewers for their comments. In the following we provide our response.

R1 (#46):
"...how robust the method is regarding the value of the parameters"
We should emphasize that we put very minimal effort into the selection of the parameter values during the experiments. Despite this fact, the behavior of our method was consistent in all the tests we conducted. For instance, using exactly the same values as in the paper, we recently performed additional experiments (included in the final paper) with the extended Middleburry stereo database (37 stereo-pairs) and we obtained very similar performance, thus further verifying the robustness of our approach. In fact, had we tried to fine-tune the parameters, it's quite likely that we would obtain further improved results.

"...Line 294 states that C was set to 10, since it gave good enough accuracy (implying that the value was found looking at the test data)"
We are referring to the accuracy on the training set. Line 294 will be corrected to read: "The linear classifiers gave good enough accuracy during training while also being fast enough to evaluate at test time".

We thank the reviewer for the minor typos.

R2 (#48):
"...The paper does not have sufficient theoretical and empirical evidence to support the argument that the heuristic works"
The Middlebury dataset is the most commonly used MRF benchmark (employed by most of existing MRF methods) and is well-established. Based on this dataset, we provide (both in the paper and supp. material) detailed empirical results on a range of different problems (stereo-matching, optical flow and image restoration) as well as on many different test images, demonstrating in all cases that our method not only significantly speeds-up the MRF optimization, but also provides most of the time a better solution than a full direct optimization.

Nevertheless, to further address the reviewer concern we will also include in the final version our recent experiments on the extended Middlebury stereo database (37 stereo-pairs). In addition, if the reviewer wishes to indicate a particular set of problems on which to test our method, we would be glad to add this experiment too.

"...The type of MRFs which may benefit from the method is limited to the ones whose solutions are piecewise smooth..."
Indeed, our approach is better suited to MRFs that exhibit locally smooth MAPs. However, we note that these are currently by far the most commonly used MRFs in computer vision, and a large set of vision problems require estimating solutions of this type.

Furthermore, such problems often involve the use of large label sets, and so being able to efficiently deal with the resulting large scale optimization tasks is currently a very important issue. Therefore, we believe that the significant speed-up and the overall flexibility of our learning-based method can potentially have an important impact on a broad range of vision applications. Moreover, to the best of our knowledge, no such learning-based approach has been explored in the context of MRFs before, and so our method can help spark further research in this area.

In addition, by adapting the meaning of the grouping function g and the used features, our approach could potentially be adjusted to other MRFs as well.

"part of the paper is not clear enough" ("...what "active" means...", "pruning matrix updating...not clear enough"), "...sec 1 is too long..."
Active means that the label is retained during optimization (and not pruned). We will add this clarification to the paper. We will also add a toy example with a pyramid construction to more clearly illustrate the pruning matrix updating. To that end, sec 1 will be reduced by removing redundancies between lines 75-110.

"...fig. 2 not very informative"
Fig. 2 shows the percentage of active labels per pixel & per scale, and aims to illustrate the behavior of pruning both
(1) across scales (demonstrating that a large part of labels become progressively non-active, which is important for the speed-up)
(2) and across space (showing also that homogeneous MAP areas are further pruned compared to MAP areas with discontinuities, which makes legitimate the learning of two different linear classifiers)

"...3 algorithms are compared, but only one curve shown in fig 1"
The horizontal axis in fig 1 represents the pruning aggressiveness factor \lambda.
a. The multi-scale approach without pruning corresponds to \lambda=inf.
b. Our multi-scale approach with pruning is shown for various values of \lambda.
c. Finally, the plotted curve represents the ratio between the currently attained energy and the energy by the direct optimization of the full MRF at the finest scale.
This allows us to compare the three algorithms with only one curve. We will edit the text to avoid the present confusion.

"The energy ratio is between the current energy and the lowest computed energy by any strategy, why is it less than 1 in many cases?"
There is a typo in the description of the experiments. Line 354 should read "We assess the performance by computing the energy ratio, i.e., the ratio between the current energy and the energy computed by the direct optimization". Hence, a ratio lower than 1 indicates that our method achieves a better solution than the traditional direct optimization (which, as can be seen, happens quite often).

R3 (#8):
"Can the proposed method be applied to graph structure with no apparent regularity?"
Yes, our method is agnostic with respect to the graph structure because both the model-coarsening and upsampling operators (sec 3.1) can be defined for an arbitrary graph. All that is needed is the definition of the grouping function g, which makes no assumption about the graph structure. This will be added to the paper.

"Could this method be used inside a learning procedure?"
We have not yet explored this option, but we thank the reviewer for this interesting idea.